# Geographical Juxtaposition: A New Direction in CPTED

**Paul Cozens** [1,*] **, Terence Love** [2] **and Brent Davern** [1]

[1]   School of Design and the Built Environment, Faculty of Humanities, Curtin University, GPO Box U1987, Perth, WA 6845, Australia

[2]   Design Out Crime and CPTED Centre, 14 Michael St, Beaconsfield, WA 6162, Australia

[*]   Correspondence: p.cozens@curtin.edu.au; Tel.: +61-89-266-7174

**Abstract:** This paper explores Oscar Newman's Defensible Space (1972) concept of geographical juxtaposition (GJ) highlighting a significant lack of research within the criminological literature over the last 50 years. We argue the concept is a key foundation in understanding crime and crime prevention theories and in developing crime prevention strategies. Findings from a systematic review of the literature are presented to illustrate the paucity of research into geographical juxtaposition. We develop and extend the concept of geographical juxtaposition beyond that originally coined by Newman to include all immediate, local, distant, and remote environmental (physical) factors. Additionally, we demonstrate, by reference to practical criminological situations, the significant and extensive role of our revised concept of geographical juxtaposition. In particular, we point to the way that focusing on geographical juxtaposition identifies serious problems in many taken-for-granted assumptions in planning theory and practice. In exploring the concept of geographical juxtaposition, we highlight ten ways it can affect crime risks and six ways using geographical juxtaposition can benefit efforts to apply crime prevention through environmental design (CPTED) more successfully when conducting a crime risk assessment. Finally, this paper briefly discusses four new CPTED principles, which emerge from our exploration of geographical juxtaposition. We identify new classes of CPTED methods and new ways of analyzing crime and offer the basis for new criminological theories.

**Keywords:** Crime Prevention through Environmental Design (CPTED); geographical juxtaposition; new ideas; new directions

## 1. Introduction

Manipulating the design, management, and use of the built and natural environment to manage risks (including crime) is not new. Indeed, a statute from the King of England in 1285 stated; "*It is likewise commanded that the highways from market towns to other market towns be widened where there are woods or hedges or ditches, so that there be no ditch, underwood or bushes where one could hide with evil intent within two hundred feet of the road on one side or the other side . . . And if perchance there is a park near the highway, it will behove the lord of the park to reduce his park until there is a verge two hundred foot wide at the side of the highway, as aforesaid, or to make a wall, ditch or hedge that malefactors cannot get over or get back over to do evil*" (Anderson et al. 2013, p. 701). This statute suggests authorities were clearly aware of the potential influence on local crime risks, of land uses in the surrounding environment.

Crime prevention through environmental design (CPTED) is increasingly popular throughout the world (Cozens 2016; Armitage and Ekblom 2019). Jeffery (1971) originally coined the phrase but acknowledged CPTED is based on Oscar Newman's *Defensible Space* (1972) ideas, rather than his own (Jeffery 1999). CPTED is subject to continuous refinement and review (Cozens et al. 2005; Ekblom 2011;

Armitage 2014; Cozens and Love 2015; Cozens 2014, 2016; Armitage and Ekblom 2019) and is Fan evolving theory and practice, located within the broader school of environmental criminology.

This paper focuses on the serious lack of attention to the concept of geographical juxtaposition (GJ); the fourth concept identified by Newman, in his book, *Defensible Space: People and Design in the Violent City* (1972). For Newman (1972), GJ is the surrounding proximal environment of a crime location. For Newman, GJ was limited solely to adjacent or proximal land uses in the immediate area. The authors, however, point to the value of the concept of GJ for understanding a much wider range of crime issues stemming from influences at a variety of levels of remoteness from the crime location. For example, at the local level, the presence of a high school influences crime risks at a local corner shop, while at a more distant scale, illegal drug production in rural areas influences the urban drug supply.

It is useful at this point to consider the First Law of Geography (Tobler 1970, p. 236) that "everything is related to everything else, but near things are more related than distant things" and the Second Law that a "phenomenon external to an area of interest affects what goes on inside" (Tobler 1999, p. 87). Supporting this Groff and McCord (2011) and others have found that locations closer to a crime generating facility will be impacted to a great degree than those farther away. Recently, in criminology and other fields, there has been more attention to this geographical aspect (e.g., Sui 2004, p. 269; Bottoms 2014, p. 1943). We suggest the situation in criminology and CPTED is more complex than this simple vector weighted approach and this is the basis for the new developments we propose in what follows.

First, however, the authors draw attention to the lack of literature and research on GJ and provide evidence of this via a systematic review of the CPTED literature. Given this shortcoming, we re-inspect and revisit the concept of GJ. The review of the literature and criminological theories and evidence leads us to develop and refine Newman's original concept and propose three new categories of GJ and strands of CPTED theory and research in relation to GJ. We explore its importance and relevance to CPTED and environmental criminology for the 21st Century. Armitage (2018) has argued CPTED needs to continue to evolve. This paper seeks to contribute towards this evolution. The outcome of such analysis, however, shakes some of the foundations of CPTED itself.

## 2. Geographical Juxtaposition in CPTED and Planning

In addition to the criminal justice system, Morgan and Homel (2013) identify two further approaches to crime prevention, the social (human) and the environmental (physical). The social approach is focused on the social and economic causes of crime and on minimizing the supply of offenders. The environmental approach modifies the physical environment to reduce opportunities for crime and includes situational crime prevention and CPTED. Ekblom (1997) Conjunction of Criminal Opportunities (CCO) helps conceptualize offender and situational perspectives and highlights the importance of "remote" causes and "immediate" precursors of crime. We can locate situational crime prevention (SCP) (Clarke 1997), as referring to those factors acting AT the scene of the crime. In contrast, GJ by its definition, compromises factors that originate OUTSIDE the immediate crime location.

At this point, before proceeding further, we would like to draw attention to what we identify as a significant, and often overlooked, contradiction in Newman's writing in which, structurally, he erroneously regards GJ as a part of his defensible space concept. In general, Newman (1972) defined defensible space concepts as being situational considerations *immediately* located (i.e., without anything in between) AT the crime location. The factors he referred to in the original concept of GJ are instead located at some distance OUTSIDE the immediate crime location. This can be seen, for example in the way that Newman (1972, p. 50) highlighted "the influence of geographical juxtaposition with 'safe zones' on the security of adjacent areas: mechanisms of geographical juxtaposition—the effect of location of a residential environment within a particular urban setting or adjacent to a 'safe' or 'unsafe' activity area".

Essentially, Newman's description of his concept of GJ was limited in special terms to the factors influencing crime from areas OUTSIDE the crime locations but PROXIMAL to it. Therefore, as the first

contribution of this paper, we draw attention to the above contradiction in Newman's writing on GJ. We refer hereafter to this amended version of Newman's ideas as Proximal GJ.

We ourselves identify and define three additional forms of geographical juxtaposition, which we refer to, following systems analysis conventions, as Micro GJ, Meso GJ, and Macro GJ. We locate Newman's Proximal GJ as the most proximal aspect of the Meso GJ category as shown in in Table 1.

**Table 1.** Four Categories of Geographical Juxtaposition.

| Geographical Juxtaposition (GJ) | Location | Comments |
|---|---|---|
| Micro GJ | GJ factors acting AT the crime location. | For example, situation crime prevention measures. For brevity we describe this in detail elsewhere. |
| Proximal GJ | GJ factors acting from locations proximal or contiguous to the crime location. | Newman's proximal geographical juxtaposition concept from now on corrected in its relationship to defensible space. For example; an alcohol serving premises located near a residence that would potentially attract or generate crime locally. |
| Meso GJ | GJ crime factors originating in areas from proximal to the crime location to physically most distant to the crime location. | Effects ranging from the above alcohol serving premises, to distant factors such as a nightclub area in a city that is on a direct railway connection to a station in a residential suburb could influence crime in that residential suburb. |
| Macro GJ | GJ factors act as remote influences on crime regardless of the location of their origin in terms of physical distance from the crime location. | For brevity we describe this in detail elsewhere. |

The crime prevention literature can be mapped onto four quadrants of a diagram (see Figure 1) along axes from "immediate" to "remote" and between social (human) and environmental (physical).

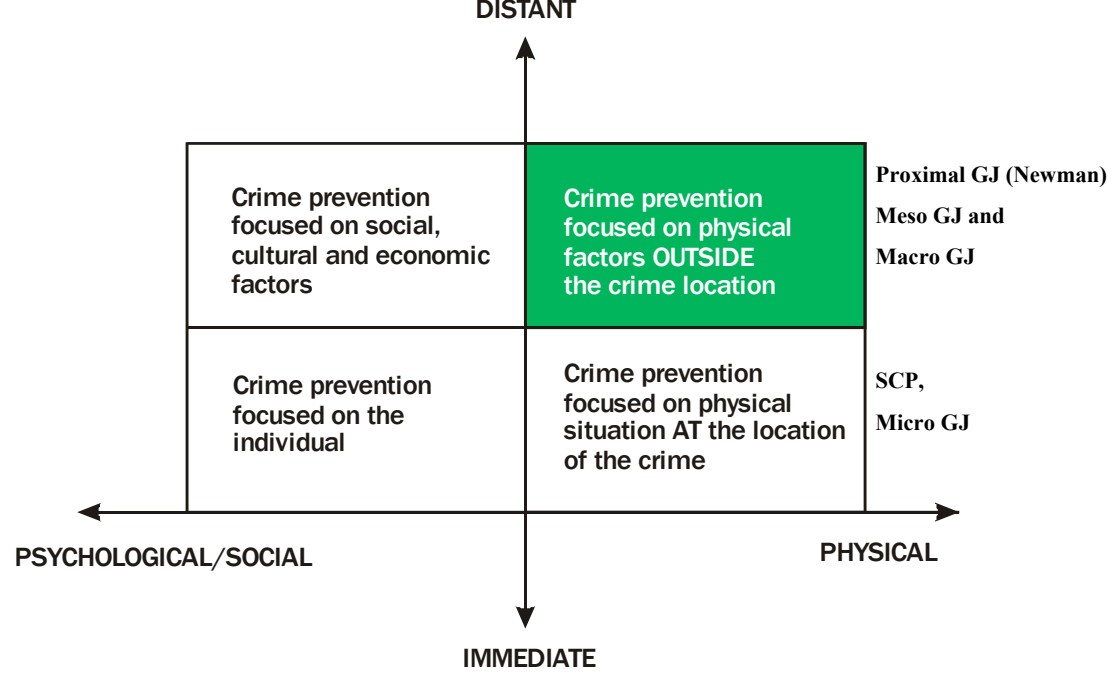

**Figure 1.** Geographical juxtaposition and the crime prevention literature.

There is extensive crime prevention literature in all quadrants in Figure 1, except the shaded, top-right quadrant. This is the area in which there is a lack of theory and literature about how local, distant and remote factors in the physical realm influence crime and CPTED undertaken at specific sites beyond the immediate and situational.

The concepts and criminological considerations of GJ primarily occupy the top-right shaded quadrant of Figure 1. This quadrant has not received much attention in the CPTED literature. This lack of attention includes both Newman's proximal geographical juxtaposition concept and the new areas of geographical juxtaposition that we define in terms of new micro, meso, and macro geographical juxtaposition categories.

The literature on the *proximal* and *meso* environmental factors in theorizing about crime and crime prevention and devising crime prevention strategies are limited and primarily include the effects of land use on crime. However, this research is not well known outside of environmental criminology. It is certainly not familiar to planners, urban designers, or architects. Crucially, their understanding of CPTED is dominated and limited by the erroneous assumption that busy places are always safe, due to the increased potential for what Jacobs (1961) called "eyes on the street". This problem is even more significant, given the international policy support for high-density, mixed-used developments in the USA, the UK and in Australia (American Planning Association 2007; DETR 1998; Commonwealth of Australia 1995). Indeed, the potential for planning to exacerbate crime as an unintended consequence has been discussed elsewhere (e.g., see Cozens 2015a, 2015b).

Jacobs (1961) work influenced Newman. She was aware of the wider influences on crime. For example, she suggested an important question about any street is "How much easy opportunity does it offer to crime?" (Jacobs 1961, p. 33) and commented "bars, and indeed all commerce, have a bad name in many city districts precisely because they do draw strangers, and the strangers do not work out as an asset at all" (Jacobs 1961, p. 41). However, she did not explicitly refer to GJ.

It is suggested that if certain types of land use are associated with certain types of crime, then other land uses may also be affected. Furthermore, this geographic/spatial influence may also affect the potential effectiveness of existing 1st Generation CPTED elements of territoriality, surveillance, image management, access control, activity support, and target hardening.

We suggest that considering the wider construct of GJ in the way that is presented here leads to:

- Identifying new ways external factors influence crime at a location.
- New theories about geographical/spacial influence on crime.
- A reviewed and refined understanding of the functioning of traditional CPTED approaches.
- A challenge to several planning archetypes and their role in crime prevention.
- Improved links between urban planning, CPTED, and environmental criminology.

## 3. Systematic Review of Literature Including the Concept of Geographical Juxtaposition

It has long been argued that GJ has been ignored within the CPTED field (e.g., Cozens et al. 1999; Cozens 2014, 2016; Cozens and Linde 2015; Cozens and Love 2017). As part of the research leading to this paper, a systematic review was recently undertaken to test the claim that GJ has not received widespread attention and to identify any research, which may (or may not) have been conducted recently.

A search of the literature of *Crime Prevention Through Environmental Design* (CPTED), *Design Out Crime*, or *Secured by Design* was carried out on 423 scientific journal articles, scholarly books, conferences papers, government reports, dissertations, implementation manuals, interview transcripts, and parliamentary reports published between 1968 and 2019. Databases used included A+ education (informit), ABI/Inform Collection (ProQuest), BioMed, BUILD CINCH, Emerald, Google Scholar, JSTOR, ProQuest, SAGE, Taylor and Francis, Web of Science, CSIRO.

The search terms used to identify literature referring to environmental/physical criminogenic factors were "geographical juxtaposition", "land use", "crime attractors" and "crime generators". These terms were chosen based on the authors' extensive experience as reviewers of the CPTED literature (e.g., Cozens et al. 2005; Cozens 2011, 2016; Cozens and Love 2015). This approach follows Sampaio and Mancini (2007) and produces a summary of the evidence in relation to a specific topic via an explicit and systematic search procedure (see Table 2).

**Table 2.** Search Terms for the Literature Review.

| Terminology | Occurrence |
|---|---|
| Geographical juxtaposition | 14 items (3.31% of all literature reviewed) |
| Crime attractors/generators | 39 items (9.21% of all literature reviewed) |
| Defensible Space | 127 items (30% of all literature reviewed) |

This review also categorized each publication according to the level of discussion, as either "terminology briefly mentioned", "moderate discussion of concept" or "extensive, in-depth discussion".

Mentioned in only 3.31% of the reviewed literature (14 publications), the limited reference to GJ indicates the paucity of focus on the wider environmental (physical) factors influencing crime beyond the immediate scale. Significantly, most of this (57%, *n* = 8) is linked to the work of Cozens and colleagues. The other six publications only briefly mention GJ. In contrast, a search on the term "defensible space", a concept of immediate (proximal/adjacent) crime prevention, occurred in over 30% of the literature.

This, together with the lack of CPTED principles and methods addressing the local, distant and remote scales of the environmental/physical approaches to crime prevention indicates that the shaded sector (see Figure 1) has been overlooked.

We suggest this lack of attention has many adverse implications. Furthermore, we suggest that when considered with a similar level of attention to that given to the other three sectors, the evidence relating to this area challenges many taken-for-granted assumptions of CPTED including those associated with urban consolidation, mixed-use developments, permeable streets, and New Urbanism.

Newman was acutely aware his ideas about crime risks from geographically juxtaposed land uses may not be well received. He stated " . . . of all the defensible space mechanisms recommended, these last two; the design of the image of the residential environment and its juxtaposition with other activities in the urban setting, will prove most offensive to architects and planners" (Newman 1972, p. 115). We wonder whether this partially explains the significant lack of research into the concept since 1972.

## 4. Literature on Geographic Juxtaposition Since 1968

Given the small number of studies (3%, *n* = 14), which have made reference to GJ, it is necessary to review some of the literature on crime and proximal/meso environmental factors—essentially, research on the effects of land use on crime.

In a paper on risky facilities, Bowers (2014, p. 390) commented "that land use and crime are inextricably linked is both intuitively plausible and well-evidenced". Interestingly, Anderson et al. (2013, p. 707) noted in the USA, before Jane Jacobs, and as early as the 1920s "proponents believed that commercial activity facilitated crime and should take place outside residential areas."

In 1999 (p. 2), Jeffery stated that CPTED ideas can be applied—but need to be carried out "within the framework of total urban planning". He continued "It does little good to target harden a convenience store located in a major urban area, while ignoring the development of a major highway a block away, or a large low-cost housing development several blocks from the store (Jeffery 1999, p. 2)". Clearly, Jeffery recognized the importance of GJ to CPTED.

The concept of GJ was brief discussed (Cozens et al. 1999, p. 256) in a paper exploring new-build housing and crime, which stated "Newman's final element concerns enhancing safety by locating the development in 'functionally sympathetic' areas, therefore making geographical juxtaposition an important, if currently vaguely conceptualized, design issue". The paper argued that the concept of GJ has important implications for new housing developments, particularly in relation to proximal land uses and the potential displacement of crime. This formed the basis for a new integrated model of defensible space developed by Cozens et al. (2001) that includes geographical juxtaposition (see Figure 2). This was based on and extends Moffat's earlier model in the same style (Moffat 1982).

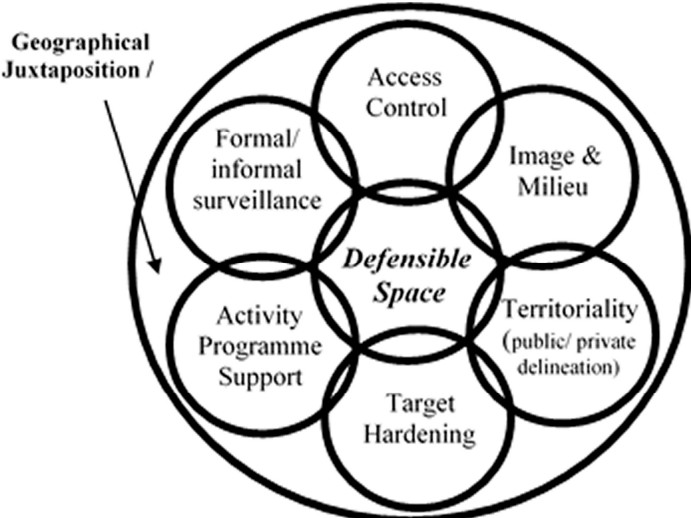

**Figure 2.** Cozens' Defensible Space Model (2001) and reference to geographical juxtaposition.

In a review of the literature on the evidence for CPTED, the model (see Figure 3) was refined to place CPTED at the center and used the term "wider environment", rather than GJ (Cozens et al. 2005). However, the authors did not provide any analysis or inspection of any research findings associated with the concept. The paper only briefly focused on the need to monitor the wider environment and the possible knock-on effects of CPTED initiatives.

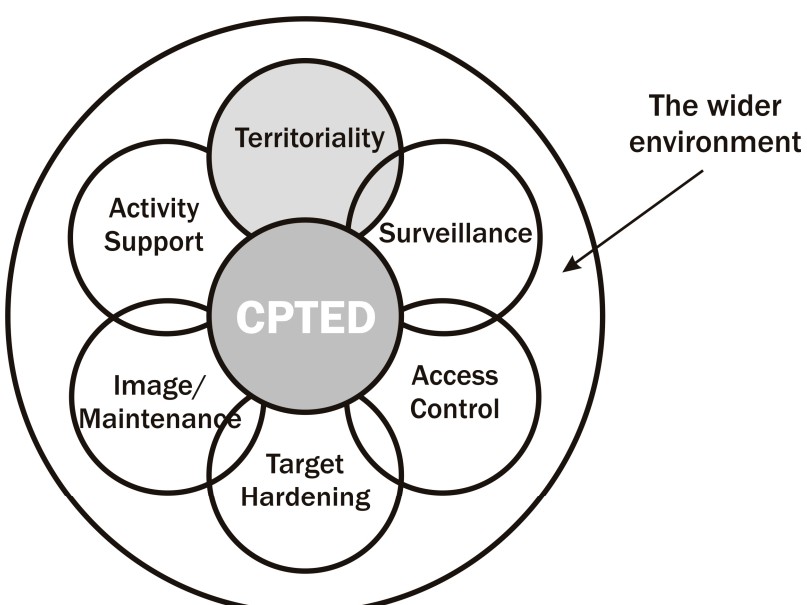

**Figure 3.** A Revised CPTED Model (Cozens et al. 2005).

In a review and critique of assumptions about land use and crime (Cozens and Hillier 2012), the importance and relevance of GJ was restated as needing consideration; however, the topic was not discussed in any detail.

A later iteration of the above model of CPTED concepts has GJ clearly located outside the other concepts in perhaps the first detailed discussion of geographical juxtaposition (see Figure 4). Links with environmental criminology, crime generators, attractors and detractors are made, and the topic is identified as a future issue for development (Cozens 2014, 2016). The model was based on an earlier analysis of the link between Newman's concept of geographical juxtaposition and the environmental

criminology land use and crime concepts of crime attractors, crime generators and crime detractors (Cozens 2011).

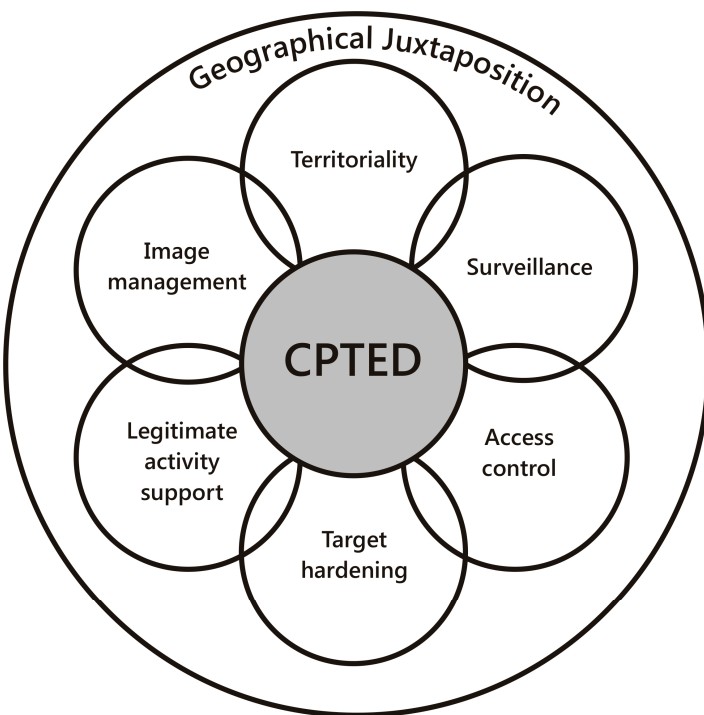

**Figure 4.** A Revised CPTED Model (Cozens 2016).

Since 2014, subsequent publications (e.g., Cozens and Linde 2015; Cozens 2016; Cozens and Love 2017) have continued to highlight the lack of attention to GJ and further explored the potential importance of this concept including "CPTED surveys need to include some kind of measurement of geographically juxtaposed land uses" (Cozens and Linde 2015, p. 86). Indeed, GJ is already being used as an additional set of criteria for conducting CPTED street audits (Cozens and Babb 2018). It has also been observed that some of the ideas within GJ appear to have been incorporated into the concept of activity support—although the details have not been discussed (Cozens and Love 2015). The theoretical links between GJ and Crime Pattern Theory (CPT) and Routine Activity Theory (RAT) have also been alluded to. Indeed, it has been argued; "geographical juxtaposition offers a powerful unifying idea that brings together several different explanations of how crime and the environment are related" (Cozens 2016, p. 55). Cozens and Love (2017, p. 6) briefly state the importance of GJ " . . . in spite of its relevance and practical significance for designing effective CPTED interventions, [GJ] has been insufficiently emphasized within the CPTED literature".

## 5. Description of Practical Examples of GJ and Significance to Crime Prevention

Mixed-use development is a very broad church and its composition can vary enormously. For example, mixed-use, or land-use heterogeneity can be a neighborhood with two land uses where there is 95% residential and 5% retail. It can also exist where there are four land uses (residential 25%, retail 25%, industrial 25%, and transportation 25%) and there are obviously many more possibilities (Wo 2019).

There are two contrasting theoretical explanations and perspectives on the relationship between mixed use and crime. One view, based on no crime evidence, is that a mixture of different land uses encourages more diversity of use and provides more "eyes on the street", which can act to reduce crime. This perspective derived from Jacobs (1961) and although unproven and unjustified by evidence has been at the forefront of modern inner city and urban consolidation policies being promoted throughout

the world. The opposing view is that mixed uses reduce the extent that citizens perceive spaces as their own and levels of informal social control are lower, increasing crime since the capacity to identify and challenge strangers is diminished. This perspective is derived from Newman (1972) but was held much earlier, in the USA in the 1920s (Anderson et al. 2013).

The crime evidence, however, independently of the above theoretical stances, indicates that for residents, crime rates are higher than they would be if the residents were living in a wholly residential area. For businesses, crime rates are lower than they would be in a wholly commercial area.

Anderson et al. (2013, pp. 711–12) provide a detailed investigation of the evidence, claiming "in general, research has concluded that contrary to Jane Jacobs's suggestion, commercial uses are associated with increased, rather than decreased crime ... and considerable research ... has shown that homogenous residential neighborhoods have lower rates of crime than mixed-use neighborhoods".

In terms of the research into the relationship between land use(s) and crime, studies have found different types of land use are associated with elevated crime risks. Overall, it is assumed non-residential land uses generate more crime than residential areas, largely as a result of reduced levels of informal social control (Stucky and Ottensmann 2009).

Some of this crime increase problem was intuitively acknowledged by UK and Australian planners in previous planning approaches that segregated residential and commercial areas.

Liquor stores are perhaps the most studied land use linked to increased crime risks. A large percentage of assaults and robberies occur inside bars (Frisbie et al. 1977). Street blocks containing bars (Roncek and Bell 1981) and liquor stores (Roncek and Maier 1991; Rengert et al. 2005; McCord et al. 2007) experienced more crime than blocks without them. Assaults and violent crime were associated with the density of retail liquor stores (Gruenewald 2011) and wine stores (Grubesic and Pridemore 2011). Livingston (2011) reported a spatial relationship between domestic violence and liquor stores and Day et al. (2012) found a positive correlation between the number of serious assaults and the distance to liquor stores. Teh (2008) reported more violent and property crime near liquor stores. Groff and McCord (2011) found exposure to bars was positively associated with violent crime and property crime, as well as with disorder.

Transit-related land uses have been linked to increased crime risks (Groff and Lockwood 2014; Kondo et al. 2016; Ridgeway and MacDonald 2017) and subway stations have been found to attract street robberies (McCord and Ratcliffe 2009). Phillips and Sandler (2015) show how public transport can influence the geographical distribution of crime. The presence of bus stops in the area (Loukaitou-Sideris 1999; Brantingham and Brantingham 1995; Newton and Bowers 2007) have been linked to increased crime risks such as burglary and robbery. Specific bus routes have higher crime risks than others (Tompson et al. 2009). Transport land uses therefore act as nodes and paths. Groff and Lockwood (2014) report a positive associated between subway stations and violent crime, property crime, and disorder.

Many studies have found retail land uses to be associated with increased crime risks across a variety of scales (Bernasco and Block 2011; Boessen and Hipp 2015; Hipp et al. (2017)). Fast food restaurants (Brantingham and Brantingham 1981) and convenience stores have been shown to attract crime (Block and Block 1995). Significant correlations between money-lending facilities (e.g., pawn shops) and both property and violent crimes have been reported (Kubrin et al. 2011). Locations with pawnshops present increased opportunities to offenders gathered nearby (Roncek 1981; Eck and Weisburd 1995). Banks can also represent risky facilities (Matthews et al. 2001).

Higher rates of violent and property crime have been associated with schools in the area (Roncek and LoBosco 1983; Roncek and Faggiani 1985; La Grange 1999; Wilcox et al. 2004; Roman 2005; Willits et al. 2013; Haberman and Ratcliffe 2015). Parks and playgrounds can also act as crime attractors (Wilcox et al. 2004; Lockwood 2006; Groff and McCord 2011). One study discovered street robberies were associated with the number of hotels and motels nearby (Smith et al. 2000). A series of studies found crime rates were elevated in street blocks with public housing projects (Roncek et al. 1981; Dunworth and Saiger 1994; McNulty and Holloway 2000).

The non-use of land is also associated with increased crime risks. Skogan (1990) has referred to derelict and abandoned buildings as a contagion. Studies show vacant land and vacancies (Spelman 1993; Ellen et al. 2013; Lacoe and Ellen 2015) can attract crime. Furthermore, stores near vacant lots have been linked with more armed robberies than stores near commercial lots (Duffala 1976).

Research has also looked at multiple (mixed-use) commercial land uses and crime. Kinney et al. (2008) studied assaults, motor vehicle thefts, and land uses, finding the distribution of land uses influenced where and when these crimes occurred. Sherman et al. (1989) identified a department store, discount store, convenience store, and bar as commercial hotspots for crime. There is extensive research associating mixed use with elevated crime risks (e.g., Luedtke 1970; Dietrick 1977; Greenberg et al. 1982; Greenberg and Rohe 1984; Taylor et al. 1985; Wilcox et al. 2004; Yang 2006). In a review, Savage and Souris (2008, p. 9) "it was striking to discover how consistent the findings indicate that mixed-land use . . . is associated with higher levels of crime".

Chang (2011) investigated burglaries, finding single housing and commercial buildings exhibited increased risks. Anderson et al. (2013) studied 205 high crime blocks in Los Angeles and reported three key findings. First, areas with residential and commercial uses exhibited lower crime than commercial only areas. Secondly, they found crime rates were lowest in residential only areas. Finally, where zoning changes added residential forms to and area, crime reduced more than in places that did not change. They concluded; "Jacobs had it backwards; rather than commercial uses reducing crime in residential areas, we found the converse to be true—residential parcels appear to reduce crime in commercial areas" (Anderson et al. 2013, p. 756). A study by Sypion-Dutkowska and Leitner (2017) found some land-use types attract crime within short distances while others deter crime. Yue et al. (2017) studied crime and 22 land-use types, finding increased risks for bike theft, burglary, and robberies were associated with convenience stores, banks, restaurants, and government facilities. They also observed how different land-use types were associated with different types of crime.

Bernasco and Block (2011) investigated crime risks and many land uses, finding blocks with crime generators/attractors within their boundaries exhibited increased risks for robbery. This also applied to bars and clubs, fast food restaurants, liquor stores, groceries, petrol stations, laundromats, pawn shops, and general stores. The effect was also found to reach adjacent blocks but decayed with distance away from the land use. They observe the 80:20 rule and how a small percentage of facilities will generate most of the crime risks. However, Bernasco and Block (2011, p. 392) suggest "it is the busy nature of facilities in general and the busy context in which facilities are often situated, rather than the facility type itself, that generates crime".

Researchers have also explored risky facilities in terms of how far their criminogenic influence extends. Kumar and Waylor (2003) investigated multiple crime types finding the effects of alcohol-related facilities decays with distance. Ratcliffe (2011, 2012) found violent crime concentrates and extends about 400 feet from bars. Groff and Lockwood (2014) examined land uses (bars, halfway houses, drug treatment centers and subway stops and schools) and crime at various distance thresholds (400, 800, and 1200 feet) and controlled for socio-demographic variables. They found subways and bars were associated with violent crime, property crime, and disorder at all distances, decaying gradually. Schools were strongly associated with increased disorder. Subway stations were particularly criminogenic and exposed the surrounding environment to increased crime for up to 1200 feet (366 m) away. SooHyun and Lee (2016) report the influence of a particular land use on local crime extends to around 400 feet (around 122 m).

Crime Pattern Theory (CPT) also relates to paths and edges. Although there is significantly less research on these elements, criminological studies have also supported and continue to underpin the theory. Wilcox and Eck (2011, p. 475) note the literature strongly indicates " . . . many facilities provide criminal opportunity and it is the contextual clustering of public-use facilities, especially along or near major roads, that is related to area crime". In terms of research on edges a study carried out

by Brantingham and Brantingham (1975) indicates that criminality predominates on the periphery of neighborhoods.

Clearly, we are beginning to understand some of the complexity of the relationship between land use and crime. However, in their review of the evidence Anderson et al. (2013, p. 727) suggest research on land use and crime needs to be more empirical and that "there is a limited understanding of the specific micro-level details of these relationships". It is suggested that CPTED audits conducted at this scale of analysis could provide interesting insights in this regard.

The evidence on land use and crime risks has focused largely on existing urban environments. Significantly, new developments and land uses in an existing area may both impact on crime and be influenced by it (see Figure 5).

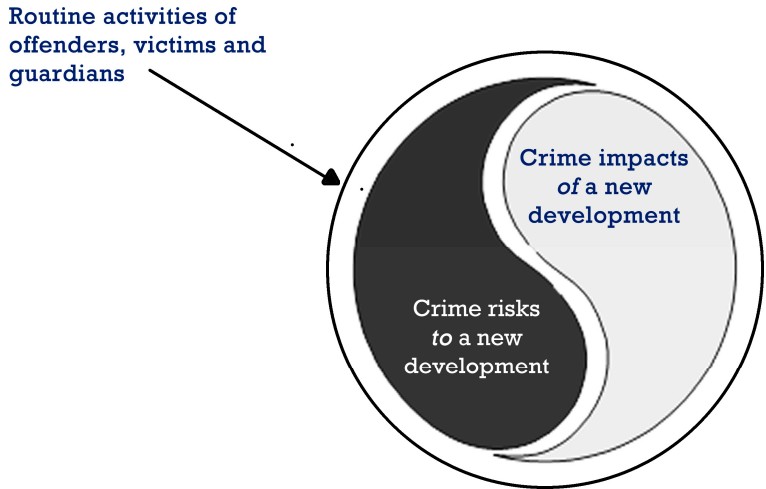

**Figure 5.** Crime Risks *to* and Crime Impacts *of* New Developments (Cozens 2011).

There may be crime impacts *of* a new development and crime risks *to* it (Cozens 2011). Crime risks *to* a new development refers to the criminogenic potential of the existing environment around it. This includes land uses and activities, which might generate or attract crime, which could affect the crime risks of the new development. For example, a new retail store could be affected by crime risks associated with several bars and nightclubs, which are located nearby. Crime impacts *of* a new development refer to the land use and activities associated with it—and how these risks might affect the community around the new development. For example, a new nightclub will create crime risks, which may impact on the safety and security of the surrounding local environment.

The relevance of GJ is therefore even more germane, given the urban consolidation and densification of our urban areas and cities across the world. In response to the perceived over-simplification of CPTED and a lack of consideration of crime risks, CPTED was redefined as "a process for analyzing and assessing crime risks in order to guide the design, management and use of the built environment (and products) to reduce crime and the fear of crime and to promote public health, sustainability and quality of life" (Cozens 2016, p. 10).

## 6. Geographical Juxtaposition, Routine Activity Theory, and Crime Pattern Theory

Cohen and Felson (1979) RAT along with Brantingham and Brantingham (1981) CPT have been cornerstones of the criminological theory underpinning CPTED. Both RAT and CPT, however, are strongly dependent themselves on GJ.

In the case of RAT, evidence indicates that crime is strongly dependent on the routine activities of criminals, victims, and guardians. These routine activities primarily exist and have their reasons for existence at a distance from the crime location, i.e., the crime comes about from the geographically based juxtaposition of the routine activities of criminals, victims, and guardians. The geographical juxtaposition of high crime neighborhoods and the density of local offenders will also be important in

assessing crime risks at a specific location. Offender travel research assumes "the density and location of crime opportunities and the various ways offenders interact with their environment strongly affects offender travel behavior" (Ackerman and Rossmo 2015, p. 238). Theories on offender location argue criminals commonly commit offences at locations that are within relatively short distances (within one mile) of where they live (Turner 1969; Stephenson 1974; Costello and Wiles 2001; Gabor and Gottheil 1984; McIver 1981; Phillips 1980; Rengert et al. 1999; Rossmo 2000; Ratcliffe 2011, 2012). RAT is clearly dependent on GJ, but has not to date, included it, referenced it, or indicated this dependence.

We take a similar position on CPT. Crime at activity nodes, paths, and edges are all issues of geographical juxtaposition. The awareness spaces of offenders, victims, and guardians are also influenced by GJ in terms of their residential location, their activity nodes, paths, and edges associated with their activities and movement. Crime generators, attractors, detractors, radiators, absorbers, and reducers are all GJ issues, which affect local crime risks.

Any and all situational patterns that influence crime must also be located in a larger environment in which they influence, and are influenced by, criminogenic factors in that larger environment, with the subsequent crime rates being influenced by the GJ of the CPT effects.

Including GJ within both CPT and RAT will certainly be challenging. Alternatively, we might also consider the contention that RAT and CPT could be regarded as sub-theories of a larger Geographical Juxtaposition Theory of Crime. This suggests a radical refinement of key opportunity theories and the need for more knowledge transfer between and across the fields of environmental criminology, CPTED and urban planning/urban design.

To further demonstrate the importance of GJ, we highlight ten ways GJ can influence local crime risks as set out in Table 3.

**Table 3.** Ways Geographical Juxtaposition May Influence Local Crime Risks.

| Influence on Crime and Public Order | The Potential Influence of GJ |
|---|---|
| 1. Dilution and concentration | The crime rate for any location may be affected by the activities in the environment nearby. |
| 2. Malign or benign displacement of crime and crime | Crime prevention efforts in a location can displace crime in ways that are more harmful or less harmful. They can also diffuse the benefits of reduced crime nearby areas and vice versa. |
| 3. Behavioral modification | The culture and behaviors of people in one location can influence nearby locations. For example, when behaviors acceptable within a pub are extended to nearby streets. All environments have local cultural cues and social structures that shape the behaviors of both law-abiding individuals and potential criminals, and this effect extends to nearby locations. |
| 4. Motivation/demotivation | The social and physical characteristics specific to a location (or environment) shape the feelings and motivations of individuals at that location. These feelings and motivations in turn, shape their behaviors. |
| 5. Distribution of crime opportunities | Criminal opportunities are increased or decreased by geographically juxtaposed features. For example, burglary opportunities may be increased by the availability of cars in a nearby un-surveilled car park. There is both an additional opportunity for cars to be stolen and an additional opportunity for transport that can increase the volume of stolen goods that can be taken. |
| 6. Nodes acting as crime attractors, generators, detractors, facilitators, enablers, precipitators, absorbers, radiators, and crime reducers | Crime risks in a location can be affected by activity nodes in the nearby environment. For example, crime risk in low crime locations may be raised by crime attractors nearby such as alcohol outlets, brothels, or a transport hub. The presence of crime attractors should influence the choice of CPTED interventions and would likely require additional CPTED interventions compared to the location with an absence of nearby crime attractors. |
| 7. Density of offenders | The number and density of offenders who live nearby or who have easy transport access from afar may affect crime rates at a location. |
| 8. Paths and accessibility | Paths, roads, rail, and other travel routes affect the accessibility to a location and the crime rate. |
| 9. Edges | Boundaries (physical/symbolic) of geographically juxtaposed areas affect crime risks as multiple criteria apply at the same location and this results in reduced informal social control, increased variety of crime risks and increased variety of crime opportunities. |
| 10. Presence of capable guardians | Land uses and local population demographics in geographically juxtaposed areas may influence the number and density of capable guardians available at any location. |

Assessing crime risks before implementing CPTED solutions has always been part of the CPTED process (Crowe 1991) but this important stage is often ignored. Most CPTED guidelines do not make any reference to assessing crime risks. The need to consider CPTED as a process, as originally orchestrated by Crowe (1991), has been continually restated (e.g., Atlas 2008; Cozens 2016; Cozens and Love 2017).

## 7. Geographical Juxtaposition and Crime Risk Assessments

GJ can improve the process and outcomes of crime risk assessments (CRAs). Assessing crime risks is therefore much more than a simple evaluation of the presence or absence of design features that promote territoriality, surveillance, image management/maintenance, access control, activity support, or target hardening. There are clear and obvious benefits of using GJ when conducting a CRA:

- GJ provides a more complete understanding of the potential sources of crime risks nearby;
- It provides a more justifiable means to help identify the types of crime most likely to occur when crime data for a site is either not available or is less than robust;
- It provides a basis to identify the most appropriate boundaries to use for a CRA, since these may not always be the same as the physical site boundaries;
- It enables the identification of feedback effects between the site and the surrounding environment that may increase or reduce crime risks;
- It provides insights and guidance to help identify which CPTED methods are likely to be most appropriate and effective, and;
- It helps to identify whether it is more effective to implement CPTED methods in the surrounding environment as well and/or instead of to the site/location to reduce crime risks in the site.

## 8. Geographical Juxtaposition and Positive and Negative Crime Feedback Loops

In any specific location, crime and crime prevention (in this case CPTED) exist as a complex system. This implies the potential for feedback loops that can either enhance the action of the CPTED activities or increase the risks of crime.

GJ exposes the existence and nature of various feedback loops in any crime/CPTED. Crime prevention using CPTED and similar strategies is not simple. Fundamentally, CPTED and its applications map directly onto what are in the Systems field are called complex socio-technical systems. One characteristic of such systems is that they always have feedback loops. Surprisingly, discussion of the role and importance of feedback loops in CPTED is almost completely absent from the CPTED and environmental criminology literature. In part, this is because the structure of the most commonly used concepts of CPTED are limited to the extent that they preclude any discussion of the important roles of feedback in causing and preventing crime. As we have developed and extended it, the concept of geographical juxtaposition provides a basis for including, analyzing, and developing CPTED methods to include feedback effects at the micro, meso, and macro scales. GJ has a strong influence on the scale of crime risks and on CPTED feedback effects.

Every characteristic location exhibits a distinctive pattern of routine activities and perceived opportunities for crime. Large public libraries, for example, are all associated with similar routine activities and perceived crime opportunities. Likewise, residential suburbs and night time economy (NTE) entertainment districts have characteristic routine activities and criminal opportunities. These shape the types of crime in and around these land uses/facilities.

When two different kinds of land use are situated next to each other, the routine activities and perceived crime opportunities of each permeate across the boundaries—in both directions. This has multiple effects. In the short term, it may increase the number of types of crime in each land use in ways that differ from those expected for each location considered individually without having regard for GJ. In criminology, this is called the "boundary effect". In parallel, these factors initiate protective effects to reduce crime—different in both land uses, yet related. Over time these factors change both

contexts in terms of their routine activities and criminal opportunities, which in turn changes the scale of offending and number of different crime types found. This feedback process of crime and CPTED changes and outcomes can be positive or negative and can be slow or fast moving or even exponential as in the case of neighborhood decline/collapse. Such GJ analyses can also expose the ways poorly implemented CPTED strategies can influence the scale and types of adverse effects on a site and its surrounding environments.

The point we are making is that analysis using GJ helps reveal the feedback loops and thus offers a better basis for understanding the criminological dynamics and for devising better CPTED solutions. The GJ of the site and its environment in terms of crime risk factors and CPTED methods can result in positive or negative feedback. This feedback can in turn significantly increase or decrease crime rates over time.

Positive crime risk feedback leads to increased crime rates over time. It occurs when criminogenic factors (crime risks) from the environment increases the crime rates and risk factors at the site, which in turn increase crime and crime risks in the environment, which then affect the site crime risks.

Similarly, positive feedback of successful CPTED can lead to decreased crime rates over time. This occurs when reductions in crime from the CPTED site also reduces the crime motivations in the nearby environment, which in turn help reduce crime risks on the site. This has also been referred to as the "halo effect" and "benign" displacement.

Negative crime risk feedback tends to lead to stabilization of crime rates. It occurs when criminogenic factors (crime risks) from the environment increase the crime rates and risk factors at the site, but at the same time the characteristics of the site are opposite and tend to reduce crime and crime risk factors in the environment—or vice versa. This is a common phenomenon and explains why crime rates tend to remain steady over time.

Negative feedback involving CPTED between site and environment works similarly. The result is outcomes that are less than what would be expected. In this situation, increased CPTED resources will need to be committed to achieve the intended outcomes.

Where feedback effects are found, it may be more effective to also apply targeted CPTED methods in the surrounding environment as well as at the site.

## 9. Geographical Juxtaposition and CPTED Methods

Above we have shown how GJ provides an essential and overarching foundational explanation of all crime and CPTED theories. For example, the theft of a purse requires the geographical juxtaposition of the crime opportunity (the purse) with a person intending to steal the purse. Similarly, natural access control comprises the geographical juxtaposition of a psychological/habitual barrier between a potential criminal and a target. We also argue that GJ provides an improved explanation of all CPTED methods as shown below in Table 4.

**Table 4.** Geographical Juxtaposition and other CPTED concepts.

| CPTED Concept | How GJ Provides a Simpler Explanation |
| --- | --- |
| Territoriality | Territoriality comprises the geographical juxtaposition of the psychological signs of potential defenders between a potential criminal and a target. |
| Surveillance | Surveillance (and sousveillance) comprise the geographical juxtaposition of potential guardians between a potential criminal and a target. |
| Image Management | Levels of maintenance and repair send the message that the space is cared for and that crime/unwanted behaviors will not be accepted. It provides a geographical juxtaposition of the owners/managers of a space into that space in front of potential offenders. |

**Table 4.** *Cont.*

| CPTED Concept | How GJ Provides a Simpler Explanation |
| --- | --- |
| Access Control | Access control is based on the existence and separation of the geographical juxtaposition of two different kinds of spaces: (a) safe and secure spaces with legitimate, private activities with legitimately owned resources; and (b) spaces with higher motivations for crime and risks that threaten to exploit the legitimate activities and resources of the former. Natural access control can be seen as the geographical juxtaposition of a psychological/habitual barrier between these two kinds of spaces, i.e., between a potential criminal and a target. |
| Activity Support | Increased levels of legitimate uses are encouraged so that their geographical juxtaposition potentially reduces crime rates nearby. A historical example is the tradition of locating churches and places of worship in higher crime environments to encourage "better" behaviors nearby. |
| Target Hardening | As for access control, target hardening is based on the existence of and the separation of the geographical juxtaposition of two different kinds of spaces: (a) safe and secure spaces with legitimate, private activities with legitimately owned resources; and (b) spaces with higher motivations for crime and risks that threaten to exploit the legitimate activities and resources of the former. Target hardening can be seen as the geographical juxtaposition of a physical barrier between these two kinds of spaces that has high costs to cross, i.e., highly secure doors between a potential criminal and a target. |

## 10. Four New Principles of CPTED

Our insights from exploring the benefits of an increased focus on GJ within CPTED and environmental criminology have helped to identify four new principles for CPTED;

- GJ is an essential basis for, and explanation of, ALL crime and crime prevention factors;
- CPTED investments should be inversely proportional to GJ factors at a distance;
- The benefits of distance from GJ factors can be achieved by obscuring the perception of criminal opportunities, and;
- The CPTED principle of natural surveillance can be divided into two parts that include promoting visibility of criminal acts and the obscuration of crime opportunities.

## 11. Geographical Juxtaposition is the Basis of ALL Crime and Crime Prevention Factors

Without geographical juxtaposition, crime does not occur. This applies from the micro to the macro scales across both physical and virtual environments. For example, at the micro scale, theft of a wallet depends on the geographical juxtaposition of the wallet's location and the hand of the person intent on stealing it.

At the most extreme situational crime prevention macro-scale, crime risks at a location depend on the geographical juxtaposition of remote conditions and remote-based potential criminals to commit crime at that location. In the virtual worlds of cyber-crime, crime risks depend on the virtual geographical juxtaposition of the attacker's code with the victim's computer data.

## 12. Make CPTED Investment Inversely Proportional to the Distance of GJ Factors

Increased attention to the GJ factors indicates that each kind of GJ factor not only influences the crime rates of locations at a distance, the effect of the GJ factors reduces with distance in a characteristic manner—different for each GJ factor. ALL crime risk factors are essentially GJ crime risk factors.

At any location, the overall crime risk is the sum of the GJ crime risk factors acting at that location. Clearly, from the above, the nearest GJ factors have a proportionally bigger effect than those further away. However, a stronger GJ factor further away may have more influence than a weaker GJ factor whose source is nearer.

The amount of CPTED investment that can be justifiably invested in reducing crime at any location is always assessed in terms of the overall crime risks at that location. Since crime risks at a location

comprise of the sum of the GJ crime risks from the environment, and the criminogenic effects of each individual GJ crime risk is inversely proportional to the distance between the location and the origin of the crime risk, then the justifiable investment on CPTED will be inversely proportional to the distances from which those crime risks geographically originate. The practical benefits in terms of CPTED planning and urban design decisions will emerge in time as research identifies the exact profile by which the effects of geographical factors diminish with distance and what combinations of uses represent what Newman (1972) called "functionally sympathetic urban areas".

## 13. Surveillance Obscuration and Crime Opportunities

The influence of geographical factors on crime risks at a location depend on the crime opportunities of that location being able to be perceived by potential criminals, primarily during their routine activities. These activities reduce with distance from their center. This has two explanations. One relates to personal interest and the second is simply a matter of geometry. Individuals are typically more interested in events closer to home (their center of routine) and the further an individual travels from their center, the larger the area is which needs to be explored at that distance. This is a squared effect such that at twice the distance from home, the area to be explored is four times bigger. At three times the distance it is eight times bigger and, at four times the distance it is 16 times bigger, and so on.

Criminals' perception and identification of crime opportunities reduces with distance for the same reasons. Individuals have insufficient time and resources to fully identify all the crime opportunities at distance, compared to locally. In effect, many crime opportunities are "obscured" from being perceived by distance of an individual from their home base. Viewing crime protection in this light points to the benefits of obscuration as a way of artificially creating geographical juxtaposition "distance". Practically, this is well known. Two examples are provided. First, hiding valuable items in a car from sight significantly reduces their risk of theft—even if the effort and risks involved in stealing them are the same as if they are in full sight. In effect, their geographical distance has been increased. Secondly, in New York, there was a large government gold bullion vault in a basement of the World Trade Center. Knowledge of this was extremely limited until after the terrorist attacks (https://www.nytimes.com/2001/11/01/nyregion/a-nation-challenged-the-vault-below-ground-zero-silver-and-gold.html). Its obscuration led to the belief that the bullion was held at a "distance" and hence the GJ influence of New York criminals was weaker.

## 14. Revision and Extension of the CPTED Principle of "Natural Surveillance"

The above points to a radical refinement in understanding relating to the CPTED principle of natural surveillance and includes two elements. The first is where the traditional CPTED concept of natural surveillance is to promote visibility and public surveillance of potentially unlawful activities in urban space. The second is the idea that natural surveillance can, at the same time, obscure the view of potential crime opportunities and crime targets. The example of hiding a laptop in the boot of the car instead of it being visible on the front seat is not hardening the target per se. It is about obscuring the crime target using GJ, which effectively removes it from view by manipulating opportunities for surveillance. This representsa new redefinition of natural surveillance for the CPTED community which will hopefully help foster research, debate, and further discussion.

## 15. Conclusions

The field of CPTED has become increasingly well-adopted throughout the world by urban planners, urban designers, architects, community safety officers, security, police, and counter-terrorism professionals. The successful adoption of CPTED to improve societies and minimize crime depends on continually updating the theories and principles in response to evidence, new knowledge, new technologies, and the development and evolution of the CPTED discipline. CPTED continues to evolve.

The evolution of CPTED has seen the development of 2nd Generation CPTED (Saville and Cleveland 1998; Saville and Cleveland 2008) and 3rd Generation CPTED (UNICRI 2011;

Mihinjac and Saville 2019). Both seek to improve CPTED as a theory by going back to its foundations in Jacobs (1961); Jeffery (1971) and Newman (1972) to re-inspect and re-define CPTED. A detailed discussion is outside the scope of this paper, but forthcoming papers will explore the importance of GJ to both 2nd and 3rd Generation CPTED. In short, this paper argues GJ is fundamental to the process and application of CPTED and in fact, provides a foundational explanation for all forms of CPTED and all its inter-related concepts.

One of the most significant and recent changes in CPTED involves designing interventions that are dictated by crime risks and the contexts of a location (Cozens 2016; Cozens and Love 2017). This is despite the fact that Crowe (1991) originally stated the need for assessing crime risks before implementing CPTED solutions. Crowe (1991, p. 35) stated "at least five basic types of information [need to] be collected and used" to make informed decisions. Later, Crowe (2000, p. 6) wrote, CPTED "is a process and not a belief system". Assessing crime risk is the first part of the process. Others have highlighted this issue more recently (e.g., Atlas 2008; Cozens 2016).

It is a challenge that many of those with the responsibility for implementing CPTED remain focused on what are now out of date CPTED design guidelines. These are largely a "one-size-fits-all" checklist of potential CPTED solutions, which are presented in the absence of any assessment of local crime risks or analysis of the local environmental context.

Another major change in CPTED is its focus on evidence-based principles, designs, and interventions. This requires CPTED designs to be justifiable by research evidence and by data on the local context and conditions.

Further change has come from ongoing critical review of CPTED theories and practices leading to development of new CPTED theories and principles (some of which are described above) and an increased focus on the crime-related conditions of each location and its environment as the basis for the detailed targeting of CPTED designs to reduce crime (Cozens and Love 2017). This has also been reflected in calls for closer links between urban planning, CPTED, and environmental criminology (e.g., Brantingham and Brantingham 1995; Cozens 2011, 2016). In addition to these recent evolutions of CPTED, this paper has provided a new and emerging direction for CPTED by highlighting the lack of attention to GJ and its potential to improve the application of the process of CPTED.

The review of 423 publications on the topic of design out crime/CPTED between 1968 and 2019 clearly reveals a dearth of research on GJ and the need to revisit and explore this concept. This research has identified four categories to extend the concept and scope of GJ. These are micro, proximal, meso, and macro GJ. Future publications will explore micro and macro GJ. In exploring the concept of GJ, we highlight ten ways it can affect crime risks and six ways using GJ can benefit efforts to apply CPTED more successfully when conducting a CRA.

We have also provided "food for thought" for architects, planners, place-makers, and urban designers, with regards to the evidence associated with different types of land uses and land-use combinations. This challenges current paradigms and planning policy in the USA, the UK, and Australia, which incorrectly assume busy places, are always inherently safe.

This paper has also reframed some of the CPTED concepts through the lens of GJ to further ground GJ as a fundamental CPTED concept. Finally, this paper has developed four new principles for CPTED, which will hopefully generate further research, discussion, and debate.

Prosocial behavior has been defined as "voluntary, intentional behavior that results in benefits for another person" (Lay and Hoppmann 2015, p. 1). Prosocial design for Armitage (2018) "recognizes that the offender is very likely to be part of our community and that perhaps enhancing an offender's emotional or moral attachment to an area may reduce their desire or inclination to commit crimes within the community". It is argued that understanding GJ and re-engaging with the concept, can help us to design, manage and use prosocial spaces more effectively and to avoid, or at least minimize, the creation of antisocial design and spaces. It can help in exploring how CPTED can be used to reduce the propensity for offending and to enhance prosocial behavior.

In summary, we conclude that the concept of GJ needs to be re-introduced as a fundamental part of CPTED theory to enable it to advance and to evolve, as Armitage (2018) suggests it must. It is opined that GJ represents the foundations to understanding and applying the CPTED process. The result is an opening up of the whole field of CPTED and the evolution of many new CPTED principles and methods.

**Author Contributions:** Conceptualization, P.C. and T.L.; methodology, P.C., T.L. and B.D., formal analysis and investigation, P.C. and T.L.; writing—original draft preparation, P.C. and T.L.; writing—review and editing, P.C. and T.L., project administration, P.C.

**Funding:** This research received no external funding.

**Conflicts of Interest:** The authors declare no conflict of interest.

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
