# Peer review of "Geographical Juxtaposition: A New Direction in CPTED"

_socsci, doi:10.3390/socsci8090252_

Round 1

Reviewer 1 Report

Dear Author,

I would really like to congratulate you for such a good piece of research. I found your work absolutely absorbing and captivating, specially the relation you make between CPTED and pro- social behaviour. I just found some editing mistakes I encourage you to amend in order to make your work brilliant:

In lines 94, 95, 99, 106 and 152, there is a weird sentence "Error!
95 Reference source not found..." In figures 2,3 and 6 the words are half-seen There is no figure 4

Again I would like to congratulate you for your work

Author Response

Thanks to the reviewers for their comments.

In lines 94, 95, 99, 106 and 152, there is a weird sentence "Error!
95 Reference source not found..." 

I don't know why / how this has happened - but I have corrected this in the revised track-changes version.

In figures 2,3 and 6 the words are half-seen There is no figure 4

All these drawings are being redrawn and saved as jpeg files. 

There is now a Figure 4 and other figures have been amended to reflect this.

Reviewer 2 Report

The ideas presented in this paper are intriguing and are definitely the "food for thought" that I think you hope they will be.  Very interesting introductions and suggested revisions to the defensible space/CPTED lexicon.

I found a few typos and other errors within the paper that should likely be addressed prior to any publication:

All Figures need to be reviewed to ensure that content is visible. 

Some typos:

Line 36 - coined THE phrase; Line 261 - acknowledged; Line 284 - convenience; Line 337 - exposed, not exposure; Line 503 - add the word of

Would like to see a bit more discussion of the impact of and risks to information presented in Figure 6. 

Author Response

Thanks to the reviewer for their comments.

I found a few typos and other errors within the paper that should likely be addressed prior to any publication:

All Figures need to be reviewed to ensure that content is visible. 

All figures are being redrawn as jpegs.

Some typos: Line 36 - coined THE phrase; Line 261 - acknowledged; Line 284 - convenience; Line 337 - exposed, not exposure; Line 503 - add the word of

All these have been corrected but are on slightly different lines (see lines 36, 271, 294, 349 and 521)

Would like to see a bit more discussion of the impact of and risks to information presented in Figure 6. 

Additional sentences have been added to clarify this - they reads as a follows:

Crime risks to a new development refers to the criminogenic potential of the existing environment geographically juxtaposed around it. This includes land-uses and activities which might generate or attract crime which could affect the crime risks of the new development. For example, a new retail store could be affected by crime risks associated with several bars and nightclubs which are located nearby. Crime impacts of a new development refers to the land use and activities associated with it - and how these risks might affect the community around the new development. For example a new night club will create crime risks which may impact on the safety and security of the local environment.